# Effect of Online Home-Based Training on Functional Capacity and Strength in Two CKD Patients: A Case Study

**DOI:** 10.3390/healthcare10030572

**Published:** 2022-03-18

**Authors:** Giulia Marrone, Elisa Grazioli, Eliana Tranchita, Attilio Parisi, Claudia Cerulli, Arianna Murri, Carlo Minganti, Manuela Di Lauro, Nicolò Piacentini, Leonarda Galiuto, Nicola Di Daniele, Annalisa Noce

**Affiliations:** 1UOC of Internal Medicine-Center of Hypertension and Nephrology Unit, Department of Systems Medicine, University of Rome Tor Vergata, Via Montpellier 1, 00133 Rome, Italy; giul.marr@gmail.com (G.M.); dilauromanuela@gmail.com (M.D.L.); npiacentini@yahoo.com (N.P.); didaniele@med.uniroma2.it (N.D.D.); annalisa.noce@uniroma2.it (A.N.); 2Department of Experimental and Clinical Medicine, “Magna Graecia” University, 88100 Catanzaro, Italy; elisa.grazioli@uniroma4.it; 3Department of Human, Movement Sciences and Health, University of Rome “Foro Italico”, 00135 Rome, Italy; eliana.tranchita@gmail.com (E.T.); claudia.cerulli@uniroma4.it (C.C.); ariannamurri@hotmail.it (A.M.); carlo.minganti@uniroma4.it (C.M.); 4Department of Cardiovascular and Thoracic Sciences, Fondazione Policlinico Universitario Agostino Gemelli IRCCS, Università Cattolica del Sacro Cuore, Largo A. Gemelli 8, 00168 Rome, Italy; leonarda.galiuto@unicatt.it

**Keywords:** chronic kidney disease, combined training, functional capacity, oxidative stress, online training protocol

## Abstract

Chronic kidney disease (CKD) is a clinical condition characterized by the loss of kidney function over time, as well as several complications affecting gastrointestinal, cardiovascular, and musculoskeletal systems. Physical exercise seems to induce positive adaptations in CKD patients, without side effects. Usually, these patients show a reduced physical activity and physical performance. The aim of this case-report was to evaluate the effects of an online training protocol on functional capacity and on muscle mass, in CKD stage III patients. Methods: Two CKD (stage III according to KDIGO guidelines) participants (1 female, Patient A; 1 male, Patient B) were enrolled and they performed an online tailored-supervised combined training lasting 12 weeks, including multi-joint strength exercises using TheraBand and an aerobic session at 65–70% of the patients’ heart rate reserve. Results: Both patients showed an improving trend on functional capacity (6 min walking test: Patient A = +3%; Patient B = +5.3%) and on strength of the upper arms (handgrip strength test-right: Patient A = +13.4%; Patient B = +19.1%; handgrip strength test-left: Patient A = +42.8%; Patient B= +12.9%), as well as a reduction in inflammation and oxidative stress biomarkers. The protocol was feasible, and no side effects were evidenced. These case studies suggest that the online combined training can produce beneficial effects in CKD patients under conservative therapy, by reducing the CKD-related complications and improving the adherence to exercise of this population of patients, overcoming logistic barriers such as transportation, availability of facilities, and working and personal-life schedule.

## 1. Introduction

Chronic kidney disease (CKD) is defined as kidney damage or a glomerular filtration rate (GFR) < 60 mL/min/1.73 m^2^ for 3 months or more, regardless of the cause [1]. It can also be diagnosed through the presence of albuminuria, detected as an albumin-to-creatinine ratio > 30 mg/g in two out of three urine samples [2]. It is estimated that 8% to 16% of the worldwide population is affected by CKD, with a higher prevalence in low- and middle-incomes countries [3]. The CKD is staged in five classes and stage III is divided into two subgroups, named IIIa and IIIb. This classification is based on estimated-GFR (e-GFR) values, obtained with the CKD-EPI formula, and on albumin-to-creatinine ratio values [4]. The primary cause of CKD can be difficult to discern and, among the main causes, arterial hypertension, diabetes mellitus, genetic diseases, primary glomerulonephritis, and congenital anomalies of the kidney and the urinary tract can occur [5]. Moreover, infections and wrong lifestyle habits seem to be additional risk factors for CKD in developing countries [1]. Depending on the stage of the disease and the patient’s clinical condition, the nutritional and pharmacological treatments are personalized [1,2,6]. Moreover, the CKD-related complications can involve several metabolic aspects and different organs or systems [7], and one of the most frequent complications, observed in these patients, is uremic sarcopenia (US). US is generally diagnosed by evaluating three parameters: muscle strength, muscle mass, and physical performance [8,9]. US induces an increased risk of falls and a greater need of hospitalization with a higher risk of mortality, compared to the general population, matched for age [10]. More in depth, US involves a series of molecular mechanisms that impair the synthesis of proteins [11]. The latest evidence reported that, alongside the conventional therapies, nutritional treatment plays a key role in the CKD management, as it counteracts the signs and symptoms related to CKD, reducing blood waste products accumulation and restoring the normal blood pH [12,13,14]. Starting from CKD stage IIIb, patients should be instructed to decrease protein intake (0.6–0.7 g *per* kg of ideal body weight *per* day) [15,16,17,18]. However, when the diet is not well-balanced and the metabolic acidosis is present, CKD patients show an impairment in body composition, with an increased risk to develop malnutrition and cachexia [19]. Alongside nutrition treatment, physical activity (PA) seems to improve several functional parameters, ameliorating patients’ quality of life (QOL) and reducing the incidence of US [20,21]. Regarding the pre-dialysis population, available studies have been conducted on small samples, and they have been limited to aerobic exercise training [22]. As reported by Kirkman et al. [23], aerobic exercise, such as cycling, walking/jogging, or elliptical activity performed for 12 weeks, 3 times *per* week, seems to improve VO2peak, and microvascular and endothelial function, ameliorating health status and reducing the cardiovascular diseases risk. Less is known about resistance training and the possible impact on overall muscular fitness, as well as on lean mass and muscle strength. Several studies have evidenced that CKD patients, to avoid muscular loss, should perform 8–10 multi-joint exercises, at least two times *per* week and with an intensity of 60–70% of 1-repetition maximum (1RM) or, as an alternative, of 5-RM [24]. To reach specific targets, such as power and hypertrophy, the strength training should include power exercises, with 6–10 repetitions performed at high velocity at 30–60% of 1-RM, and exercise for hypertrophy, performing 8–12 repetitions at slow to moderate velocity, 60–80% of 1-RM [25]. On the other hand, resistance training alone seems to have no significant effects on aerobic capacity [24]. The latest scientific studies have focused their attention on combined aerobic and strength protocols, which seem to improve the 6 minutes walking test (6MWT), sit-to-stand test, gait-speed test, and health-related QOL, as well as lipid profile and body composition, and to reduce the muscle weakness in CKD patients [26,27]. Despite encouraging preliminary results, the intensity, duration, and frequency of this type of training are still unclear.

Several barriers inhibit patients from participating in PA and exercise programs, as well as performing them for a long-term period. Those barriers (regarding prevalently hemodialysis-HD patients) include pathological comorbidities, such as chronic pain, chronic fatigue, and diabetes mellitus [28,29,30]; logistics conditions, such as transportations, availability of facilities, and work schedule [31,32]; psychological barriers such as fear of falling, lack of motivation, and poor self-efficacy [29,33]. As it was already studied in other chronic non-communicable diseases, like cancer and neurodegenerative diseases [34,35], a supervised online training could be an optimal solution to overcome those barriers, because it does not require expensive equipment and it can be easily performed at home, anytime. According to our knowledge, there are no studies about online training interventions tailored to CKD patients. Regarding these considerations, the primary aim of this case study was to evaluate the effect of an online combined exercise protocol on functional capacity and strength in two CKD patients under conservative therapy. Moreover, the body composition, QOL, and several laboratory parameters, such as inflammatory biomarkers, free oxygen radical test (FORT), and free oxygen radical defense (FORD), were analyzed in order to understand the impact of this type of training on CKD.

## 2. Materials and Methods

A 57-years-old female patient (Patient A) and a 56-years-old male patient (Patient B) were recruited in the UOC of Internal Medicine, Center of Hypertension and Nephrology Unit of Policlinico Tor Vergata University Hospital, in February 2021.

In Patient A, the primary cause of CKD was nephroangiosclerosis, diagnosed in 2005. Patient A previously underwent to surgery for a crural hernioplasty, in 1998. The patient carried out the following usual therapy: beta blockers at a dose of 5 mg/day and angiotensin II type 1 receptor blockers at a dose of 40 mg/day. The renal function laboratory parameters, collected before the intervention (T0), were: azotemia 44 mg/dL, creatininemia 1.32 mg/dL, uricemia 8.6 mg/dL, e-GFR 41.63 mL/min, and albumin-to-creatinine ratio 10.59 mg/g. The CKD stage of this patient was IIIb.

In Patient B, the primary cause of CKD was obstructive nephropathy (congenital bilateral pyeloureteral joint stenosis surgically treated in 2014). He previously underwent inguinal hernioplasty and tonsillectomy surgeries in childhood. Concerning comorbidities, he presented only the systemic arterial hypertension. The patient carried out the following usual therapy: angiotensin-converting enzyme inhibitors at a dose of 5 mg/day, alpha blockers at a dose of 2 mg/day, beta blockers at a dose of 5 mg/day. The renal function laboratory parameters, collected before the intervention (T0), were: azotemia 49 mg/dL, creatininemia 1.71 mg/dL, uricemia 6.1 mg/dL, e-GFR 41.62 mL/min, and albumin-to-creatinine ratio 48.18 mg/g. The CKD stage of this patient was IIIb.

During the period of the online training program, the two patients did not modify their usual pharmacological treatment, to avoid any possible bias on the obtained result [36,37]. 

All baseline patients’ epidemiological characteristics are reported in Table 1. Ethical approval was provided by the Ethical Committee of the Policlinico Tor Vergata University Hospital, Rome (protocol number R.S. 223/20) and all procedures met the guidelines set by the Declaration of Helsinki of 1975. The participants provided a written informed consent.

### 2.1. Intervention

Online training program: The PA protocol combined a resistance band training with mobility, balance, and aerobic training. It was performed three times *per* week, one hour (+15 additional minutes when needed) for each session, using the Microsoft Teams platform, for 12 weeks, under the supervision of a Specialized Sport Science Trainer. Patients familiarized themselves with the Microsoft platform prior to starting the intervention, and they were instructed to record their heart rate (HR) through a personal HR monitor and the intensity of the session through the BORG scale (0–10). The training program was monitored by specialized trainers. Every 4 weeks, the volume and workload were adapted to increase the exercise intensity. Each session was structured with the warmup phase: 15 min (mobility and balance). The strength phase: 20/30 min of resistance band training. During the first 4 weeks, exercises involving the anterior/posterior chain were proposed, where patients performed two sets of two circuits composed of 4 multi-joint exercises, with bodyweight or with the resistance band. Core training, which includes exercises aiming to improve abdominal, lumbar, and pelvis muscle strength, was incorporated in the session as well. In the last 8 weeks of exercises, stimulating the upper/lower chain, patients started with two sets of 4 exercises during weeks 4–8, and then in weeks 8–12, they performed three sets of 3 exercises. The adaptation was introduced gradually according to patients’ capacities. At the end of the sets, the trainer asked the patients to monitor their HR and to provide their fatigue index on the BORG scale. The aerobic phase: 10/15 min of aerobic training. During the first weeks, the training was performed without music to allow patients to learn the choreography. Once they mastered the basic movements, music was introduced. The intensity started at ≃50–60% of heart rate reserve (HRR) (≃120 bpm) in the first two weeks of training, and patients completed the training at ≃65–70% of HRR (≃128 bpm). HR and BORG were monitored at the end of this session. The cool down phase: 5/10 min of stretching of all muscle groups involved in the protocol. An example of the schedule exercise is provided in the Appendix A.

### 2.2. Assessments

Functional and psychological outcome measures were assessed at baseline and after 12 weeks of intervention. At the enrolment (T0) and after 12 weeks of training (T1), a venous blood and urinary sample was collected from the two enrolled patients to analyze blood and urinary routine parameters, inflammatory biomarkers. Moreover, we performed FORT, and FORD tests on capillary sample [38,39].

### 2.3. Functional and Psychological Evaluation

6MWT: To assess the functional capacity, the patient was asked to walk as far as possible (no running or jumping) for 6 min on a flat corridor while the researcher recorded the HR every minute, the walking distance, and the fatigue sensation, through the Borg Scale (0–10), at the end of the 6 min [40]. 

Handgrip strength test (HST): to assess the upper body strength, the patient was asked to squeeze the dynamometer (Jamar plus model) while seated on a chair with the elbow of the working hand at 90° close to the side. The trial was performed with both hands, alternately, for three times, and the average score was taken into consideration [41].

Anthropometric assessment: anthropometric parameters were detected. Body weight (kg) was measured to the nearest 0.01 kg using a balance scale (Seca 711, Hamburg, Germany). Height (m) was measured using a stadiometer to the nearest 0.1 cm (Seca 220, Hamburg, Germany). Body mass index (BMI) was calculated as body weight divided by height squared (kg/m^2^).

Body impedance analysis: to assess the body composition, both patients underwent to bioelectrical impedance analysis (BIA). Resistance, reactance, impedance, and phase angle were measured using a BIA 101S instrument (Akern/ RIL System-Florence, Florence, Italy). For the evaluation of body composition, we considered fat free mass (FFM), fat mass (FM), body cell mass (BCM), BCM index (BCMI), total body water (TBW), intracellular water (ICW), and extracellular water (ECW). Moreover, the skeletal muscle mass index (SMMI) was calculated [41].

Short Physical Performance Battery (SPPB): to assess the lower extremity functioning, this is a combination of 3 tests assessing gait speed (4 m walking), power (repeated chair stand), and balance (tandem test). Each test was scored out of 4 and their sum indicates the level of performance, where 12 is the best performance [42]. 

Sit and Reach (S&R): to measure the flexibility of the back and hamstrings muscles, it can be useful to evaluate the functional ability of the legs in terms of walking speed and dynamic balance.

Baecke questionnaire: to assess PA level, this questionnaire evaluates the level of PA in the last 12 months, investigating three different spheres: work (WA), sport (SA), and leisure activity (LA) [43].

Short Form Health Survey 36 (SF-36): to evaluate adult patients’ perceptions of their own health and well-being, this consists of 36 items concerning physical and mental health, which are measured by analyzing 8 different domains. For each section, the higher the score, the higher the health [44].

## 3. Results

Functional, clinical, and psychological data pre- and post-treatment were reported, and the differences were evaluated through the percentage variation considering the initial data as a reference. So long as this is a case study on two patients, it is not possible to generalize the results, but it provides preliminary data that need to be tested systematically on a larger number of cases.

### 3.1. Functional Evaluations

According to the functional evaluation analysis, reported in Table 2, a general improvement of the functional capacity was evidenced in both patients (6MWT: Patient A= +3%; Patient B= +5.3%). The fatigue sensation after the test decreased in both patients, suggesting a functional adaptation despite the increased distance performed in the 6MWT. This decline was more evident in the female subject (−50%) than in the male (−10%) one. The strength results of the upper body evidenced a general increasing trend in both patients (HST-R: Patient A = +13.4%; Patient B = +19.1%; HST-L: Patient A = +42.8%; Patient B = +12.9%). The SPPB evaluation showed a great starting point of Patient B, which collected the higher score possible before and after the intervention. Patient A reported a lower starting point that increased after 12 weeks of intervention (SPPB = +20%). Lastly, the flexibility evaluation, through the sit and reach test, evidenced a tendency to improve at the end of the online PA program.

### 3.2. Anthropometric Assessment and Laboratory Parameters

According to the results related to the body composition parameters (Table 3), after 12 weeks of intervention, body weight evidenced a small tendency to decrease in Patient A (−0.4%) and to increase in Patient B (+0.1%), as well as the BMI value (Patient A = −0.7%; Patient B= +0.4%). To better understand these results, FM and FFM were evaluated too. Patient A evidenced a small increase in FM (+1.7%) and a small decrease in FFM (−1.4%), while patient B showed a decrease in FM (−23.6%) and an increase in FFM (+7.6), confirming the importance of evaluating the body composition alongside the BMI value. Regarding SMMI: for Patient A, we observed a slight decrease, after the online PA, for this parameter (−5.3%); and for Patient B, we observed an increase (+15.5%).

Routine laboratory parameters (Table 4) showed some interesting variations. In Patient B, both C-reactive protein (CRP) and erythrocyte sedimentation rate (ESR), after the online PA program, were reduced (−42.8% and −10%, respectively), while Patient A showed a trend of stability. Regarding lipid profile, we observed a total cholesterol decrease for both patients (Patient A, −15%; Patient B, −18%).

Regarding FORT and FORD tests, in Patient A, T0 showed high levels of oxidative stress (OS), and after the online PA program, FORT reduced by 61.9%, while Patient B at baseline showed normal levels of OS, remaining stable. This result is related to the decrease in FORD, observed in patient A. In fact, the reduction in OS levels induced a major antioxidant defenses expenditure.

### 3.3. Questionnaires Scores

The SF-36 questionnaire (Table 5), which evaluates the general health of patients, showed positive results only in Patient B, who had improved Social Functioning (+14.2%), Pain (+12.5%), and Health Change (+50%). Patient A did not report changes in Social Functioning, Pain, and Health Change, but she presented a decreased level in Physical Functioning (−14.2%), Role Limitation Physical (−100%), Role Limitation Emotional (−50%), Vitality (−10%), Emotional Well Being (−14.2), and General Health (−25%), confirming the evidence that the perceived health-related QOL of women is poorer than that of men, and women report a higher symptom burden and greater symptom severity than men [45]. The total score of the Baecke questionnaire, which evaluates the level of PA, showed an improved level of activity after the online PA program for both in Patient A (+2.6%) and in Patient B (+6.1%).

## 4. Discussion

According to our knowledge, this model of online PA is innovative and the obtained results are in agreement with the literature data. In fact, PA in CKD patients has been demonstrated to be a useful tool to prevent CKD-related comorbidities, such as US, and to slow their progression [33,46]. In particular, the main benefits induced by PA are the lowering in systemic blood pressure, increase in VO2max, increase in muscle strength, and an amelioration of physical performance [22]. Indeed, the CKD patients who participated in our interventional study protocol showed an improvement in the 6MWT, suggesting that an online supervised training plays a key role in the upgrading of functional capacity, confirming the data reported by Pajek et al. [40]. 6MWT has been identified as a mortality predictor in end-stage renal disease (ESRD) patients, evidencing that an increase of 100 mt can be related to an improvement in life expectancy. For this reason, the increase trend of this parameter, as it has been reported in our study, could be a good starting point to lay the foundations of its adjuvant role in the clinical management of CKD patients under conservative therapy. Another simple and reliable method useful to predict the survival rate, as well as the incidence of US in CKD patients, is the HST [41,47]. In fact, low HST values are associated with an increased risk for all-cause mortality in these patients, and it seems a suitable method for the evaluation of musculoskeletal function [48]. Moreover, an inverse correlation was found between HST and malnutrition-inflammation score in CKD patients under conservative therapy [49]. According to our data, after the 12 weeks of exercise, both patients increased the HST, in both arms, suggesting a positive impact of this type of protocol on the general strength and on muscle function, as well as on US incidence. Regarding the outcomes on body weight, Patient A reported a BMI decrease, while Patient B reported a BMI increase after the training protocol. After a most accurate analysis, through the BIA, the results evidenced a FM increase and a FFM decrease in Patient A, whereas patient B showed opposite results. Several studies have reported a reverse association between BMI and the risk of death for all-causes in HD patients [50,51,52], but this index lacks in delineating FM and FFM [53], which can provide a better mortality prediction than the BMI alone, in CKD patients [54]. Interestingly, according to Pereira et al.’s [55] formula, we evaluated the SMMI by BIA, for the two patients. In particular, the obtained results showed a slight decrease for Patient A (although the SMMI was within the limit of normality in both time-points of the study) and we observed an increase for Patient B, although he did not reach the normality value at the end of the online PA program. The obtained results suggest the possible beneficial role of the online PA program for this patient population, and this hypothesis is more corroborated by the results obtained in Patient B, who increased the SMMI. For this reason, in our clinical trial ongoing, we are collecting body composition parameters such as FM and FFM through bioimpedance analysis, to better investigate the impact of PA on body composition changes in CKD patients [56]. CKD is associated with an increased OS and it has also been described by several authors as a progression factor to ESRD [57,58,59], representing a nontraditional risk factor for all-cause mortality in this population [46]. The OS is a phenomenon that involves a higher production of ROS that triggers mediators such as nuclear factor kappa-light-chain-enhancer of activated B cells (NF-κB) and tumor necrosis factor (TNF)-α, which, in turn, amplify the inflammatory status [60]. Our data confirmed that PA plays an important role in counteracting OS. In particular, in Patient A, after the intervention, we observed a decrease both in FORD and FORT, suggesting a reduction in OS. These data are interesting because the FORD test expresses the blood antioxidant capacity, and its reduction, related to FORT reduction, indicates a major consumption of antioxidant species in order to decrease OS. In patient B, we observed standard levels for both FORD and FORT, probably due to the baseline normal value of FORT. For this reason, these two tests did not reveal significant results in Patient B. In addition to FORD and FORT, we analyzed the routine inflammatory parameters such as CRP and ESR. In Patient A, CRP showed a slight increase; meanwhile, ESR did not change. These results are probably linked to the body weight. Pre-obesity and obesity are historically related to inflammatory status, and several authors speculated that higher BMI values are associated with enhanced CRP concentrations [61,62]. Furthermore, female subjects have a stronger association between BMI and inflammatory status than men [63]. On the contrary, Patient B during the study protocol showed a normal BMI value, and both CRP and ESR showed decreased concentrations, suggesting the positive role of PA on inflammatory biomarkers in the presence of normal BMI. Lastly, regarding the SF-36 questionnaire, in the literature, there are plenty of studies confirming that exercise and PA positively influence QOL in CKD patients [64]. Despite the SF-36 highest scores recorded in Patient B, the psychological results of this case report are not in line with those reported in the literature. These data are probably due to the pandemic period in which the protocol was performed, which has negatively influenced the QOL of the general population, especially subjects affected by chronic pathologies [65,66]. The questionnaire about the PA levels reported a general increase in activity degree, suggesting that this type of intervention can induce some behavioral changes in CKD patients. On the other hand, it could be the consequence that at T0 assessment (February 2021), there were many restrictions due to the coronavirus disease-19 (COVID-19) pandemic that forced people to stay at home or to avoid certain activities, thus affecting the score. Meanwhile, at T1 (at the end of May 2021), many restrictions have been suspended. However, in the exceptionality of the COVID-19 situation and of the lockdown, the online PA represented for those patients a safe opportunity to train in a supervised manner. Recent studies showed that daily PA has shrunk dramatically during the COVID-19 pandemic and the closure of gyms and sport centers highly reduced PA levels in patients affected by chronic pathologies [67,68]. An additional outcome of our interventions aimed to counter this negative trend that enhances the risk of mortality and morbidity in CKD patients [69,70]. The limitation of this study is that it was performed on only two patients; consequently, it is not possible to generalize the results, but it provides preliminary data that need to be tested systematically with a larger number of cases.

## 5. Conclusions

The tailored online exercise program was revealed to be safe, to not induce side effects, and to have a great adherence in CKD patients under conservative therapy. Moreover, the online training seems to improve several functional and clinical parameters related to the decrease in several CKD-related comorbidities, such as cardiovascular diseases or US. Lastly, the online modality could improve the adherence to exercise, overcoming logistic barriers such as transportation, availability of facilities, and working and personal-life schedule. The study provided encouraging results that should be investigated in an in-depth manner. 

## Figures and Tables

**Table 1 healthcare-10-00572-t001:** Patient A and Patient B baseline characteristics.

Epidemiological Characteristics	Patient A	Patient B
Age (years)	57	56
Height (cm)	163	184
Weight (kg)	71	83
BMI (kg/m^2^)	27.1	24.4
Gender	F	M
Stage of CKD	IIIb	IIIb
Primary cause of CKD	nephroangiosclerosis	chronic pyelonephritis
Pharmacological treatment	Beta blockers (5 mg/day), angiotensin II type 1 receptor blockers (40 mg/day)	Alpha blockers (2 mg/day), beta blockers (5 mg/day), angiotensin-converting enzyme inhibitors (5 mg/day)

Abbreviations: BMI, Body Mass Index; CKD, Chronic Kidney Disease; F, Female; M, Male.

**Table 2 healthcare-10-00572-t002:** Patient A and B functional assessment before (T0) and after (T1) interventions. The comparisons between pre- and post-intervention data are expressed by the percentage variation (%).

Functional Tests	Patient	T0	T1	%
6MWT (m)	A	500	515	+3%
	B	750	790	+5.3%
6MWT Borg	A	4	2	−50%
	B	5	4.5	−10%
HST-R (kg)	A	26.1	29.6	+13.4%
	B	51.8	61.7	+19.1%
HST-L (kg)	A	18.9	27	+42.8%
	B	47.7	53.9	+12.9%
SPPB (score)	A	10	12	+20%
	B	12	12	0%
S&R (cm)	A	3	3.5	+16.6%

Abbreviations: 6MWT, 6 Minutes Walking Test; 6MWT Borg, Borg Scale (0–10) after the 6MWT; HST-R, Handgrip Strength Test Right; HST-L, Handgrip Strength Test Left; SPPB, Short Physical Performance Battery; S&R, Sit And Reach Test.

**Table 3 healthcare-10-00572-t003:** Patient A and B anthropometric assessment before (T0) and after (T1) interventions. The comparisons between pre- and post-intervention data are expressed by the percentage variation (%).

Anthropometric Assessment	Patient	T0	T1	%
Weight (kg)	A	71.8	71.5	−0.4%
	B	83.1	83.2	+0.1%
BMI (kg/m^2^)	A	27.1	26.9	−0.7%
	B	24.5	24.6	+0.4%
FM (%)	A	34.9	35.5	+1.7%
	B	24.1	18.4	−23.6%
FFM (%)	A	46.9	46.2	−1.4%
	B	63.1	67.9	+7.6%
TBW (%)	A	47.6	47.3	−0.6
	B	55.5	61.5	+10.8%
ECW (%)	A	48.0	46.1	−4%
	B	49.8	43.4	−13%
ICW (%)	A	52.0	53.9	+3.5%
	B	50.2	56.6	+12.7%
BCM (%)	A	53.4	51.3	−2.9%
	B	56.2	49.3	−12.2%
BCMI	A	9.4	8.9	−5.3
	B	10.5	9.9	−5.7%
SMMI (kg/h^2^)	A	7.5	7.1	−5.3%
	B	9.1	10.5	+15.4%
Waist circumferences (cm)	A	92.9	93.1	+0.2%
	B	106.1	105.5	−0.5%

Abbreviations: BCM, Body Cell Mass; BCMI, Body Cell Mass Index; BMI, Body Mass Index; ECW, Extracellular Water; FM, Fat Mass; FFM, Fat Free Mass; ICW, Intracellular Water; SMMI, Skeletal Muscle Mass Index; TBW, Total Body Water.

**Table 4 healthcare-10-00572-t004:** Patient A and B laboratory parameters before (T0) and after (T1) interventions. The comparisons between pre- and post-intervention data are expressed by the percentage variation (%).

Laboratory Parameters	Patient	T0	T1	%
CRP (mg/L)	A	5.1	5.6	9.8%
	B	1.4	0.8	−42.8%
ESR (mm/h)	A	45	45	0%
	B	20	18	−10%
Albumin (g/dL)	A	4.5	4.6	+ 2%
	B	4.5	4.2	−6%
Glycemia (mg/dL)	A	89	99	+9.7%
	B	75	78	+4.2%
Total cholesterol (g/dL)	A	230	195	−15%
	B	245	200	−18%
LDL-cholesterol (mg/dL)	A	164	127	−21.6%
	B	141	161	+14%
HDL-cholesterol (mg/dL)	A	52	47	−9%
	B	58	50	−12.7%
Triglycerides (mg/dL)	A	147	131	−11.4%
	B	92	108	+17.3%
FORT (U)	A	421	160	−61.9%
	B	160	160	0%
FORD (mmol/L Trolox)	A	1.74	1.13	−35%
	B	1.01	1.09	7.9%

Abbreviation: CRP, C-Reactive Protein; ESR, Erythrocyte Sedimentation Rate; FORT, Free Oxygen Radical Test; FORD, Free Oxygen Radical Defense; HDL, High-Density Lipoprotein; LDL, Low-Density Lipoprotein.

**Table 5 healthcare-10-00572-t005:** Patient A and B Questionnaires scores before (T0) and after (T1) interventions. The comparisons between pre- and post-intervention data are expressed by the percentage variation (%).

Questionnaires Scores	Patient	T0	T1	%
SF-36 PF	A	70	60	−14.2%
	B	100	96	−5%
SF-36 RLP	A	20	0	−100%
	B	100	100	0%
SF-36 RLE	A	66.7	33.3	−50%
	B	100	100	0%
SF-36 V (E/F)	A	50	45	−10%
	B	85	80	−5.8%
SF-36 EWB	A	56	48	−14.2%
	B	88	88	0%
SF-36 SF	A	50	50	0%
	B	87.5	100	+14.2%
SF-36 P	A	45	45	0%
	B	80	90	+12.5%
SF-36 GH	A	40	30	−25%
	B	75	75	0%
SF-36 HC	A	50	50	0%
	B	50	75	+50%
BAECKE TOT	A	7.6	7.8	+2.6%
	B	8.1	8.6	+6.1%

Abbreviation: PF, Physical Functioning; RLP, Role Limitations—Physical; RLE, Role Limitations—Emotional; V(E/F), Vitality (Energy/Fatigue); EWB, Emotional Well-Being; SF, Social Functioning; P, Pain; GH, General Health; HC, Health Change; TOT, Total Score.

## Data Availability

Not applicable.

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
