# Peer review of "Effect of Online Home-Based Training on Functional Capacity and Strength in Two CKD Patients: A Case Study"

_healthcare, 2022, doi:10.3390/healthcare10030572_

Round 1
Reviewer 1 Report
Dear editors:
It is a great honor and pleasure for me to be invited as the reviewer for this important work. Giulia Marrone1 and the co-authors have evaluated the effect of online home-based training on functional capacity in CKD patients in COVID-19 era. This study topic is interesting, attributing to Prof. Parisi’s long-term efforts and contributions in this scientific field. Although the article provides some results that refresh our understanding of AKI risk factors, I have a number of comments concerning this study:
- According to the updated KDIGO guidelines, the stages of CKD patients should be presented as “GxAx”. The comparison of eGFR/ BUN/Cr/ UACR should be analyzed.
- In the abstract section, the abbreviation of “MWT” in abstract section should be spelled out.
- In the abstract section, some spaces should be inserted in the result part about Handgrip strength test (HST).
- In the Table 1, BMI “27,1” should be expressed as “27.1”.
- To identify the issue of malnutrition-inflammation complex or protein-energy wasting, the lab data of “albumin” should be added in the Table 3 and text.
- As discussed in Line 276, the bioimpedance analysis should be added to avoid confounders and confirm the improvement of uremic sarcopenia in in CKD patients.
- Concerning BMI, the parameters of metabolic syndrome, including the waist circumference, blood glucose and lipid profiles, should be provided.
- Would you please provide the website of training program for readers?
Thank you for giving me the opportunity to review this interesting article. After minor revision, this article could be considered for publication.
Sincerely,
Reviewer 2 Report
The authors present a case report regarding the impact of supervised online/home-based training for CKD patients focusing on their functional performance. The case report is interesting on the matter that it focuses on the utilization of tele-health (in the broader context of the term) tools for patient management. This tools would gain more ground on the years to come due to several reasons (covid-19 included). Some comments that would further improve the work in my opinion.
1) I would suggest to put first the intervention and subsequently the functional and psychological evaluation
2) Is the training protocol somehow validated/certified for these patients? Also, were the trainers certified for this type of exercises?
3) Line 44 for personalized approaches needs some references. Also, did the patients followed similar treatments? Maybe since it is a case report, pharmacological treatments could be placed in table 1.
4) These patients often show side effects from the treatments that receive. Any chance pharmacological treatments could impact the results or the physical and psychological evaluation? (some relative references: https://doi.org/10.3390/ijerph17239101; https://doi.org/10.7399/fh.2017.41.2.10508 etc.)
5) It would be interesting to see comparison between online and real-life physical training (even in the risk of reporting bias).
Reviewer 3 Report
In the abstract, the exercise/training program is not identified. Please insert the exercise protocol.
Percentages are not sufficient for the statistical results in the abstract. Instead, please add p-values or effect sizes, etc.
Reword lines 47-49.
Lines 49-54 needs to be removed. The information is not relevant to the focus of the manuscript. The focus of the manuscript is functional capacity, not ROS and inflammation.
Based on the aims of the case study, the title of the manuscript will need to be changed.
In the tables, replace the comas with periods.
What was the fitness level of the two individuals?
This is not a case study but more of an experimental clinical study from the methodology.
More than two individuals are needed for this type of study.
There was no statistical analysis performed or mentioned throughout the entire manuscript. Therefore, percentages are not enough to determine outcomes. Also, were the results from a clinical or statistically significant perspective?
Based on my previous comments, there is no way that the results can support the discussion.
Round 2
Reviewer 3 Report
Thank you for addressing my comments.